# Retinal oxygen supply shaped the functional evolution of the vertebrate eye

Christian Damsgaard[1,2]\*, Henrik Lauridsen[3,4], Anette MD Funder[5], Jesper S Thomsen[6], Thomas Desvignes[7], Dane A Crossley II[8], Peter R Møller[9], Do TT Huong[10], Nguyen T Phuong[10], H William Detrich III[11], Annemarie Brüel[6], Horst Wilkens[12], Eric Warrant[13], Tobias Wang[1], Jens R Nyengaard[14†], Michael Berenbrink[15†], Mark Bayley[1†]

[1]Section for Zoophysiology, Department of Bioscience, Aarhus University, Aarhus, Denmark; [2]Department of Zoology, University of British Columbia, Vancouver, Canada; [3]Department of Clinical Medicine, Aarhus University, Aarhus, Denmark; [4]Meinig School of Biomedical Engineering, Cornell University, Ithaca, United States; [5]Department of Forensic Medicine, Aarhus University, Aarhus, Denmark; [6]Department of Biomedicine, Aarhus University, Aarhus, Denmark; [7]Institute of Neuroscience, University of Oregon, Eugene, United States; [8]Department of Biological Sciences, University of North Texas, Denton, United States; [9]Section for Evolutionary Genomics, Natural History Museum of Denmark, University of Copenhagen, Copenhagen, Denmark; [10]College of Aquaculture and Fisheries, Can Tho University, Can Tho, Viet Nam; [11]Department of Marine and Environmental Sciences, Marine Science Center, Northeastern University, Nahant, United States; [12]Zoological Institute and Zoological Museum, University of Hamburg, Hamburg, Germany; [13]Department of Biology, Lund University, Lund, Sweden; [14]Core Center for Molecular Morphology, Section for Stereology and Microscopy, Centre for Stochastic Geometry and Advanced Bioimaging, Department of Clinical Medicine, Aarhus University, Aarhus, Denmark; [15]Institute of Integrative Biology, University of Liverpool, Liverpool, United Kingdom

**\*For correspondence:**
cdamsg@zoology.ubc.ca

[†]These authors contributed equally to this work

**Competing interests:** The authors declare that no competing interests exist.

**Abstract** The retina has a very high energy demand but lacks an internal blood supply in most vertebrates. Here we explore the hypothesis that oxygen diffusion limited the evolution of retinal morphology by reconstructing the evolution of retinal thickness and the various mechanisms for retinal oxygen supply, including capillarization and acid-induced haemoglobin oxygen unloading. We show that a common ancestor of bony fishes likely had a thin retina without additional retinal oxygen supply mechanisms and that three different types of retinal capillaries were gained and lost independently multiple times during the radiation of vertebrates, and that these were invariably associated with parallel changes in retinal thickness. Since retinal thickness confers multiple advantages to vision, we propose that insufficient retinal oxygen supply constrained the functional evolution of the eye in early vertebrates, and that recurrent origins of additional retinal oxygen supply mechanisms facilitated the phenotypic evolution of improved functional eye morphology.

## Introduction

The light-absorbing retina lining the back of the vertebrate eye is an outgrowth of the forebrain and shares the same high oxygen demand with other neural tissues (*Country, 2017*). Despite its high demand for oxygen, the retina of most vertebrates remains avascular (*Country, 2017*; *Yu et al.,*

*2009*) and must be supplied by oxygen diffusing from adjacent vascular structures. Thus, retinal thickness is constrained by the diffusive distance between the retinal mitochondria and the ubiquitous choroidal capillary network (choriocapillaris) behind the retina (*Country, 2017*; *Buttery et al., 1991*; *Chase, 1982*; *Yu et al., 2009*). Diffusion is governed by Fick's law, where adequate oxygen delivery to sustain a thicker retina can be achieved through reduced diffusion distance, increased surface area of retinal capillary beds, or increased oxygen partial pressure gradient. Any combination of such enhancements will increase oxygen flux and open opportunities for enhanced retinal thickness. While thickness itself may be of little adaptive consequence, its enhancement allows for changes in retinal morphology that are directly associated with visual performance. These include increased density of photoreceptors and ganglion cells within the retina (*Walls, 1937*; *Querubin et al., 2009*; *Potier et al., 2017*) as well as the stacking of rods and cones to increase light sensitivity (*Rodieck, 1998*).

Shorter diffusion distance from blood to the retina and/or larger diffusive area can be achieved by developing capillaries either within the retina itself or on its light facing side (intra- and pre-retinal capillaries, respectively). These additional capillary networks, which are analogous to those of the human retina, are found in a small minority vertebrates, including examples in teleost fish, amphibians, reptiles, birds, and particularly mammals, and are thus scattered across the vertebrate phylogeny (*Yu et al., 2009*; *Country, 2017*; *Chase, 1982*; *Meyer, 1977*; *Buttery et al., 1991*; *Walls, 1937*). While improving the oxygen supply, such retinal capillaries may interfere with the light path to the retinal cells and hence introduce a trade-off between oxygen delivery and visual acuity (*Country, 2017*; *Yu and Cringle, 2001*).

Early in the evolution of the ray-finned fishes, amino acid substitutions in haemoglobin produced an exceptional pH-sensitivity of oxygen binding called the Root effect (*Root, 1931*). The subsequent evolution of a specialised capillary bed behind the retina (choroid rete mirabile) allowed for localised reduction of pH, which promoted efficient oxygen off-loading. In combination with the counter-current arrangement of its capillaries it became possible to generate localised extreme oxygen partial pressures ($PO_2$) in excess of 1300 mmHg (*Wittenberg and Wittenberg, 1962*; *Wittenberg and Wittenberg, 1974b*; *Wittenberg and Haedrich, 1974a*) (see *Figure 3—figure supplement 1* and Interactive 3D Model for detailed vascular overview). Thus, many of the ray-finned fishes enhance oxygen flux to the retinal cells not by increased diffusion area or diminished diffusion distance, but instead by massively increasing the steepness of the diffusion gradient using the oxygen secretory mechanism.

Whilst the evolutionary relationships between the size of the Root effect and the development of the choroid rete mirabile has been studied in some detail (*Berenbrink et al., 2005*; *Verde et al., 2008*), the relationship between oxygen secretion and retinal thickness has not been examined. The same can be said of the relationship between the development of intra- or pre-retinal capillaries and retinal thickness. It has been convincingly argued that the choroid rete mirabile evolved only once in an early ray-finned fish but was subsequently lost independently within five clades of teleosts, concomitantly with a reduction in Root effect (*Berenbrink et al., 2005*). These losses allow for the investigation of whether secondary reductions in oxygen secretion resulted in thinning of the retina or in compensatory introduction of pre- or intra-retinal capillarization to maintain retinal thickness.

The present study was therefore designed to detect co-evolutionary changes between retinal oxygen supply and retinal morphology. Since the ray finned fishes are known to present all three enhancements in oxygen flux (*Country, 2017*; *Yu and Cringle, 2001*), we developed a phylogenetic model based on ray-finned fishes to illuminate the effects of enhancements and losses of oxygen secretion on retinal thickness, and we used the lobe-finned fishes as outgroup. Within these species, we measured the magnitude of the Root effect, the extent of retinal thickness, and the degree of retinal capillarization (*Supplementary file 1*). We then scanned the literature and included additional data on Root effect magnitude and the presence of the choroid rete mirabile to increase the power of the ray-finned fish model (*Supplementary file 1*). In addition, since intra-retinal capillarization is widespread among mammals, we used the model to evaluate the correlation between oxygen supply and retinal thickness in mammals based primarily on literature data.

Using this model, we tested three hypotheses on vertebrate eye evolution. Firstly, since retinal capillarization and oxygen secretion have not been reported in elasmobranchs or jawless vertebrates, we tested the hypothesis that the archetypal bony fish retina was thin and was supplied solely by the choriocapillaris. Next, we tested the hypothesis that any evolutionary enhancement of

retinal oxygen flux was associated with thicker retinae. Finally, we tested the hypothesis that modulation of oxygen secretion changes the need for retinal capillarization, such that loss of oxygen secretion should be associated with either thinning of the retina or introduction of extra capillary supply routes.

## Results

In the following, we initially analyse the evolution of retinal thickness followed by a similar analysis of the evolution of retinal oxygen flux mechanisms. Finally, we present the analysis of the evolutionary interactions between retinal morphology and its oxygen supply.

### Evolution of retinal and ocular morphology

The thickness of the retina and its layers was quantified with both in vivo high-frequency ultrasound and histology on 34 ray-finned fish species, three lungfishes and two mammals (*Figure 1*, *Videos 1–2*). This core data was supplemented with literature values of retinal thickness from 14 tetrapods. We then used a maximum likelihood ancestral state reconstruction to model retinal thickness across all branches within a phylogenetic tree encompassing these species.

This analysis suggested a thin retina (194 μm) in the last common ancestor of bony fishes some 425 million years ago (MYA), that is before the split of the ray- and lobe-finned fishes (*Figure 1*) supporting the first hypothesis. The analysis revealed that retinal thickness doubled independently on six occasions within the bony fish phylogeny. Thus, the doubling occurred once within osteoglossiform fishes, once in salmoniformes and four times in perciform fishes. In all of these groups retinal thickness exceeded 500 μm in some species (*Figure 1*). The analysis also showed that retinal thickness halved three times within the vertebrates, falling to below 96 μm within several species (*Figure 1*).

To test whether increased eye size was associated with a thicker retina, we first computed the body mass corrected residuals in eye mass (phylogenetic general least squares (PGLS), t = 12.7, p<0.001, n = 79, *Figure 2A*) and showed that these residuals were indeed strongly associated with increases in retinal thickness (PGLS, t = 21.6, p<0.001, n = 36, *Figure 2B*). This suggests that species that invest in large eyes (compared to similar sized species) also invest in increased retinal thickness.

### Evolution of additional retinal vascularization

Next, we used stereological analyses on high-resolution computed tomography scans, magnetic resonance imaging scans, and histological sections of eyes from 58 different vertebrates to identify and quantify the type and extent of capillary networks supplying the retina. In addition, we measured the Root effect in blood samples from 43 species and combined these two data sets to reconstruct the evolutionary history of the oxygen flux to the vertebrate retina (*Figure 3*). This analysis indicated that the most recent common ancestor of bony fishes was devoid of a choroid rete mirabile (probability for presence = 0.1%), devoid of intra-retinal capillaries (probability for presence = 0.4%), and likely also lacked pre-retinal capillaries (probability for presence = 32.4%) (*Figure 3*). These results strongly support the first hypothesis that the retina of this ancestor to bony fish only relied on the choroidal capillaries lining the back of the retina for oxygen delivery similarly to what is found in the extant coelacanth and Australian and South American lungfishes (*Figure 3—figure supplement 2*).

Further, the reconstruction showed that the choroid rete mirabile originated at least twice in the ray-finned fishes. Firstly, in the lineage leading to *Amia*, and secondly, in a common ancestor to the teleosts (*Figure 3*). Loss of the choroid rete mirabile actually appears to have been a common phenomenon through the radiation of the ray-finned fishes, as we find a surprising 23 losses (*Figure 3*). Intra-retinal capillaries originated independently within ray-finned fishes and mammals (*Figures 3* and *4*): Within the ray-finned fishes, intra-retinal capillarization originated at least twice, in the branches leading to the European eel, *Anguilla anguilla*, and the elephant nose fish, *Gnathonemus petersii* (*Figure 3*). Applying this model on mammalian data (*Chase, 1982*; *Leber, 1903*; *Samorajski et al., 1966*; *Moritz et al., 2013*; *Kolmer, 1927*; *Bellhorn, 1997*; *McMenamin, 2007*), reveals that the retina of the most recent common ancestor of mammals was most likely devoid of capillaries (anangiotic) (probability for presence of capillaries = 1.2%). Capillarization of the whole retina (holangiotic retina) seems to have originated from an anangiotic ancestor on at least three occasions within the mammalian phylogeny including twice within marsupials (~63 and 54 MYA) and

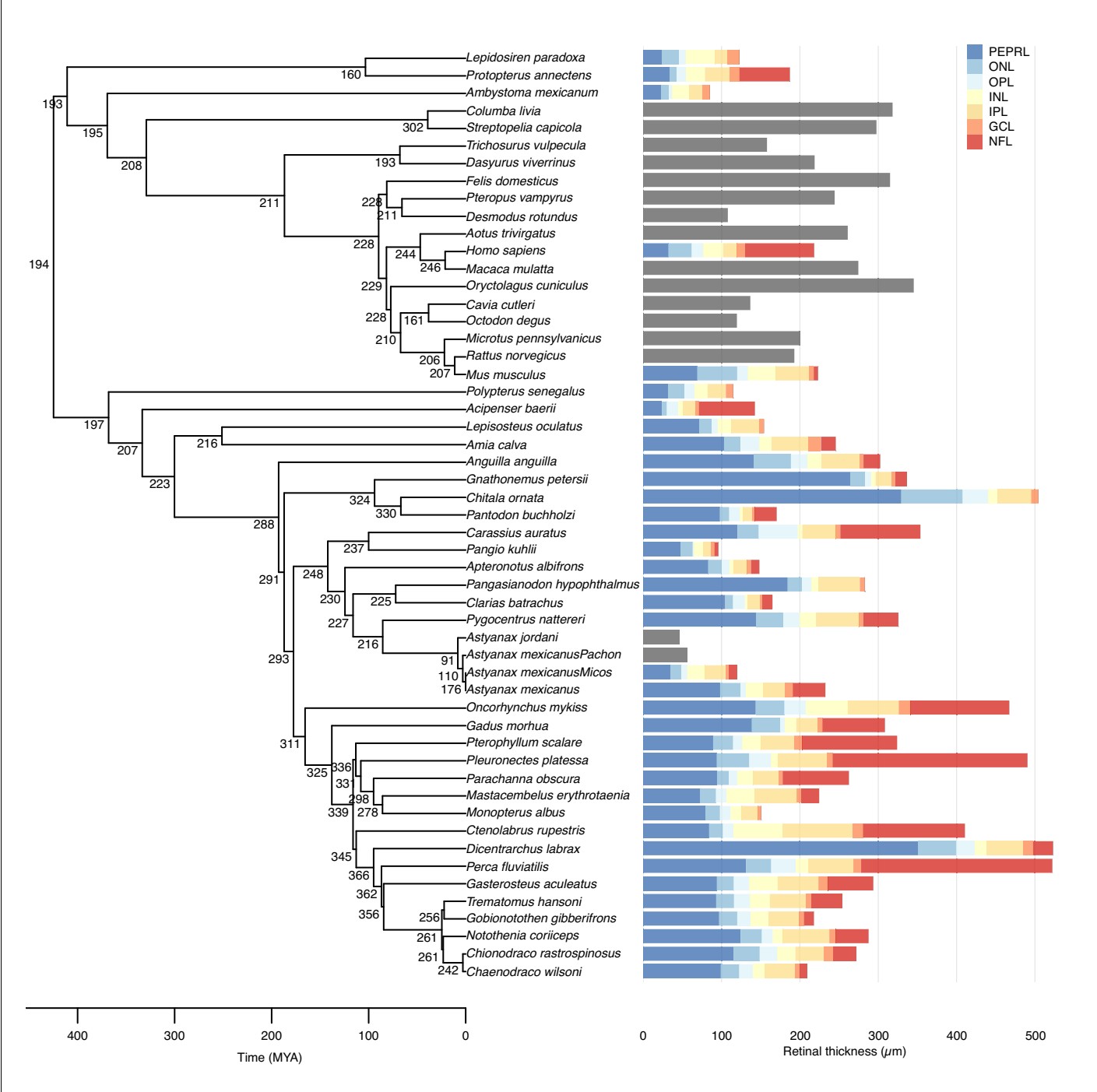

**Figure 1.** Evolution of maximal retinal thickness in 53 bony fishes. Measured (bars) and reconstructed (internal nodes) values for maximal retinal thickness plotted on a bony fish phylogeny. Stacked colours represent thicknesses of individual retinal layers (retinal thickness in species without data on retinal layers thickness are shown in grey bars). Retinal thickness was measured using in vivo ultrasound, histology, or acquired from the literature, and retinal layer thickness was measured on histological sections. Ancestral states were inferred using maximum likelihood. Retinal layer abbreviations: PEPRL, pigment epithelium and photo receptor layer; ONL, outer nuclear layer; OPL, outer plexiform layer; INL, inner nuclear layer; IPL, inner plexiform layer; GCL, ganglion cell layer; NFL, nerve fiber layer.

once in a common ancestor of the eutherian superorders Laurasiatheria and Euarchontoglires (~96 MYA) (*Figure 4*). The analysis also shows secondary reductions in retinal capillarization within mammals. Thus, there are at least five complete losses within the eutherian mammals (*Figure 4*) where

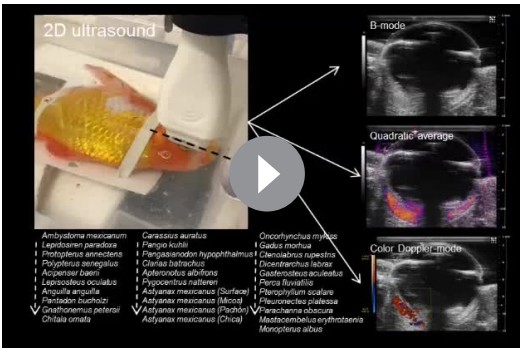

**Video 1.** Two-dimensional ultrasound videos in B-mode, quadratic average mode, and colour Doppler mode in a mid-coronal plane through the eyes of fishes. *Ambystoma mexicanum, Lepidosiren paradoxa, Protopterus annectens, Polypterus senegalus, Acipenser baerii, Lepisosteus oculatus, Anguilla anguilla, Pantodon bucholzi, Gnathonemus petersii, Chitala ornata, Carassius auratus, Pangio kuhlii, Pangasianodon hypophthalmus, Clarias batrachus, Apteronotus albifrons, Pygocentrus nattereri, Astyanax mexicanus* (Surface), *Astyanax mexicanus* (Micos), *Astyanax mexicanus* (Pachòn), *Astyanax mexicanus* (Chica), *Oncorhynchus mykiss, Gadus morhua, Ctenolabrus rupestris, Dicentrarchus labrax, Gasterosteus aculeatus, Perca fluviatilis, Pterophyllum scalare, Pleuronectes platessa, Parachanna obscura, Mastacembelus erythrotaenia,* and *Monopterus albus.*
https://elifesciences.org/articles/52153#video1

retina change from a holangiotic to an anangiotic state, as well as two partial reductions (holangiotic to merangiotic) where capillaries in the retinal periphery are lost but remain around the optical disc (*i.e.*, in the branches leading to European rabbit, *Oryctolagus cuniculus*, and musk deer, *Moschus fuscus*).

## Evolutionary dynamics in retinal vascularization in ray-finned fishes

The presence of a choroid rete mirabile is associated with an elevated Root effect. Thus, the Root effect is significantly higher in species with choroid rete mirabile compared to species without (phylogenetic analysis of variance simulation (pAOV), F = 122, p<0.001, n = 68, *Figure 5A*). Further, there is a negative correlation between the extent of pre-retinal capillarization and Root effect magnitude (PGLS, t = -4.62, p<0.001, n = 19; *Figure 5B*) supporting the idea of pre-retinal capillaries causing light scattering and thus a trade off in visual performance. This inverse relationship between retinal capillarization and oxygen secretion was supported by reconstructing retinal oxygen supply in the lines of descent connecting the most recent common ancestor of bony fishes and extant fishes exhibiting retinal oxygen secretion, such as European perch, *Perca fluviatilis* (*Figure 6A*). In this evolutionary trajec-

tory, the Root effect increased linearly over time (*Figure 6A*). When the Root effect increased above 40% around 200 MYA, the choroid rete mirabile evolved, allowing a decrease in pre-retinal capillarization (*Figure 6A*). Similar reductions in retinal capillarity in the evolutionary trajectory leading to several other teleosts with retinal oxygen secretion further supported this relationship (evolutionary trajectories to all extant species are deposited on GitHub, https://github.com/christiandamsgaard/Retinaevolution). Moreover, the reconstructed evolutionary trajectories of retinal oxygen supply indicate reverse causation between retinal capillarity and oxygen secretion during secondary losses of oxygen secretion. For example, oxygen secretion was secondarily lost in the lineages leading to European eel, *Anguilla anguilla* (*Figure 6B*), and striped catfish, *Pangasianodon hypophthalmus* (*Figure 6C*), and these losses coincided with the origin of intra-retinal capillarization and extended pre-retinal capillarization in the two respective lineages. All three of these findings strongly support the third hypothesis by showing an inverse relationship between retinal capillarization and oxygen secretion

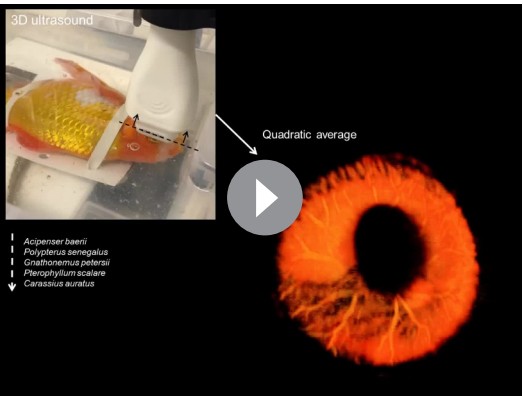

**Video 2.** Three-dimensional, quadratic-averaged ultrasound slice videos through the coronal plane of the eyes of five species with fundamentally different eye circulatory patterns. *Acipenser baerii*, Only choriocapillaris; *Polypterus senegalus*, Choriocapillaris and pre-retinal capillaries; *Gnathonemus petersii*, Choriocapillaris, pre-retinal capillaries and intra-retinal vessels; *Pterophyllum scalare*, Choriocapillaris and choroid rete mirabile; and *Carassius auratus*, Choriocapillaris, choroid rete mirabile and pre-retinal capillaries.
https://elifesciences.org/articles/52153#video2

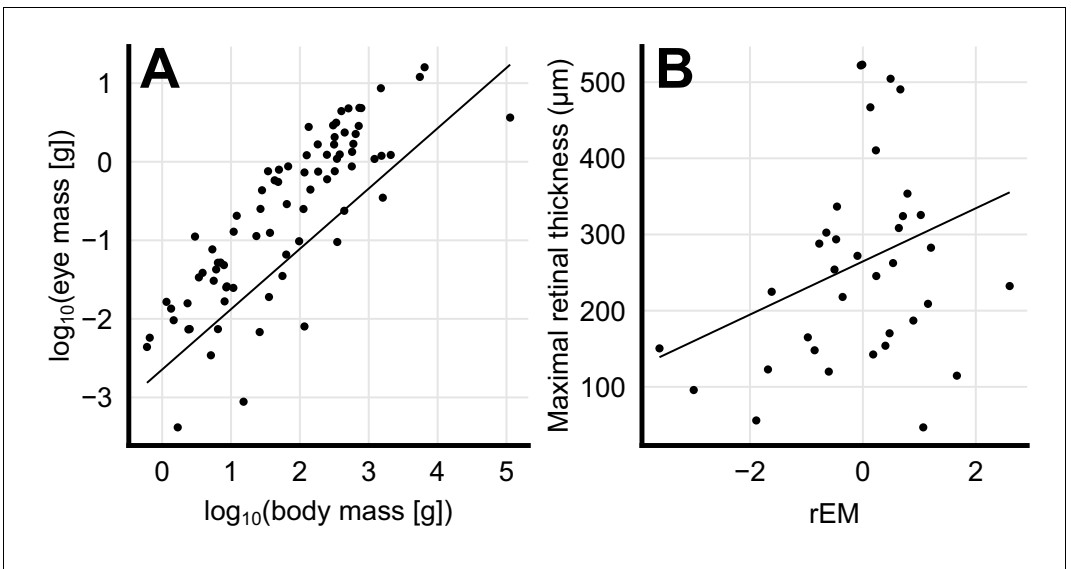

**Figure 2.** Scaling of eye mass and retinal thickness. (A) Species-mean values of eye mass and body mass (dots) showing an allometric scaling relationship. Solid line depicts a phylogenetic general least squares (PGLS) fit to the data ($\log_{10}$(eye mass [g])=0.77 $\log_{10}$(body mass [g]) – 2.64, t = 12.7, p<0.001, n = 79). (B) Positive correlation between retinal thickness and residual eye mass (rEM) that are the residuals of the PGLS fit, which are body mass-independent measures of eye mass. Solid line depicts a PGLS fit to the data (retinal thickness = 34.9 rEM + 265, t = 21.6, p<0.001, n = 36).

through the dual presence of Root effect haemoglobins and the choroid rete mirabile.

## Relationships between retinal oxygen supply and retinal thickness

The retinal thickness in the ray-finned fishes was strongly associated with the presence of a choroid rete mirabile which represents a proxy for oxygen secretion (pAOV, F = , p=0.00264; *Figure 7A*). This association was found in the evolutionary trajectory connecting the most recent common ancestor of bony fishes and extant teleosts showing a 30% increase in retinal thickness in the branch where the choroid rete mirabile evolved ~200 MYA (e.g., perch; *Figure 6A*). Conversely, in branches where the choroid rete mirabile was secondarily lost, retinal thickness decreased substantially, which further supported the tight relationship between oxygen secretion and retinal morphology (e.g., loaches; *Figure 6D*). The only species that deviated from this loss of retinal thickness downstream of loss of oxygen secretion are those in which oxygen flux was to some extent maintained by the introduction of intra- or pre-retinal capillaries that replace the loss of partial pressure gradient with an increase in diffusive surface area or reduced diffusive distance (*Figure 6C*; *Figure 7A* black vs. orange symbols, p=0.005, PGLS). This relationship can be tested in mammals, which never evolved oxygen secretion, but which show a strong relationship between retinal thickness and presence of intra-retinal capillarization (pAOV, F = , p = 0.00168; *Figure 7B*).

We also examined two groups of fishes with 'natural knock-out' of retinal morphology or haemoglobin to examine whether they follow the patterns described above. Firstly, Antarctic icefishes that lost haemoglobin could be compared to close relatives that retained haemoglobin allowing examination of how a sudden loss in blood oxygen carrying capacity affected retinal morphology. Here, the reconstruction showed that species with or without haemoglobin show remarkably similar retinal thickness (*Figure 8*). Those that lost haemoglobin and hence oxygen secretion show an extensive pre-retinal capillarization (previously described by *Wujcik et al., 2007*) allowing the maintenance of retinal thickness (*Figure 8*). This finding strongly emphasises how loss of blood oxygen supply can be compensated by increasing the diffusion area via new capillary networks. Secondly, we explored how relaxed selection on vision affected the oxygen supply to the eye. During the recurrent cave invasions by the Mexican tetra, *Astyanax mexicanus*, vision has been lost through varying degrees of eye regression (*Figure 9*). The ecotypes within this species displayed variation in Root effect magnitude and choroid rete mirabile size, where ecotypes with greatest eye regression showed lowest

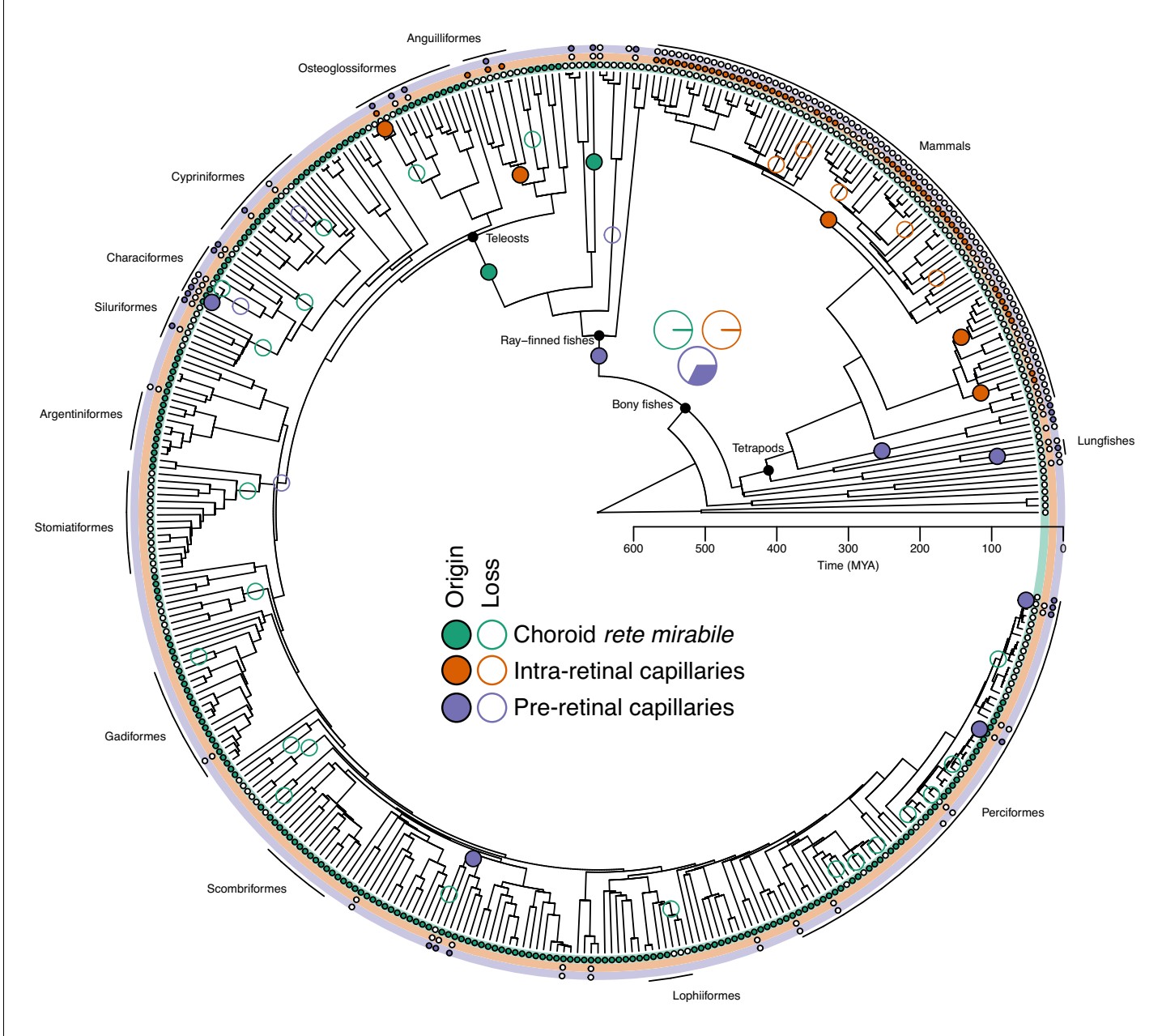

**Figure 3.** Evolution of retinal oxygen supply mechanisms in vertebrates. The time-calibrated phylogeny of jawed vertebrates plotted with the inferred evolution of additional retinal oxygen supply mechanisms using observations from this study and the literature. Labels on terminal branches indicate the retinal oxygen supply phenotype of each species, where empty and filled circles represent the absence and presence of specific mechanisms (no circle represents no information available). Empty and filled circles in internal nodes represent inferred origins and losses of these mechanisms based on stochastic character mapping. Pie charts in the centre shows the Bayesian posterior probability for the presence of the three types of capillaries in the most recent common ancestor of bony fishes. Arches in the periphery denote mammals, lobe-finned fishes, and larger orders of ray-finned fishes. See *Figure 3—figure supplement 3* for a phylogeny with species names.

The online version of this article includes the following figure supplement(s) for figure 3:

**Figure supplement 1.** Circulation in the bony fish eye.

**Figure supplement 2.** Examples of retinal blood supply types in bony fishes.

**Figure supplement 3.** Evolution of retinal oxygen supply mechanisms in vertebrates.

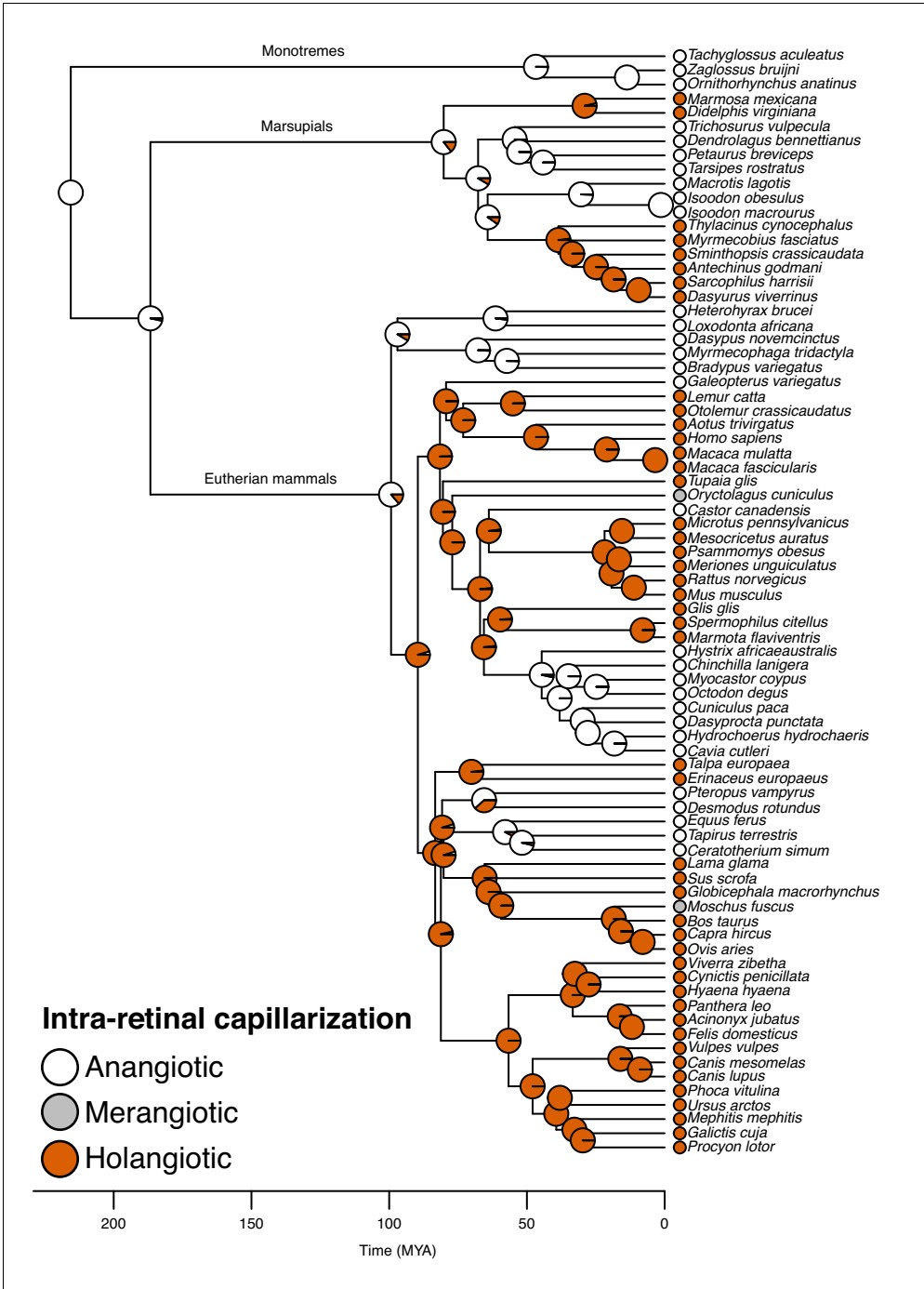

**Figure 4.** Evolution of retinal capillarization in mammals. Mammalian phylogeny showing retinal capillarization examined in extant species (tips; literature data: *Chase, 1982*; *Leber, 1903*; *Samorajski et al., 1966*; *Moritz et al., 2013*; *Kolmer, 1927*; *Bellhorn, 1997*; *McMenamin, 2007*) and reconstructed retinal capillarization (internal branches), showing of holangiotic (capillarization of the whole retina; orange), merangiotic (capillarization confined to the retina around the optic nerve; grey), and anangiotic capillarization (little or no capillarization; white). Pie charts indicate ancestral states on internal nodes showing posterior probabilities summarised from stochastic character mapping.

Root effect magnitudes as well as diminished choroid rete mirabile size (*Figure 9*). In addition, the Micos ecotype that exhibited an intermediate stage of eye regression had almost halved the retinal thickness compared to the basal surface ecotype. Thus, both of these 'natural knock-out' examples

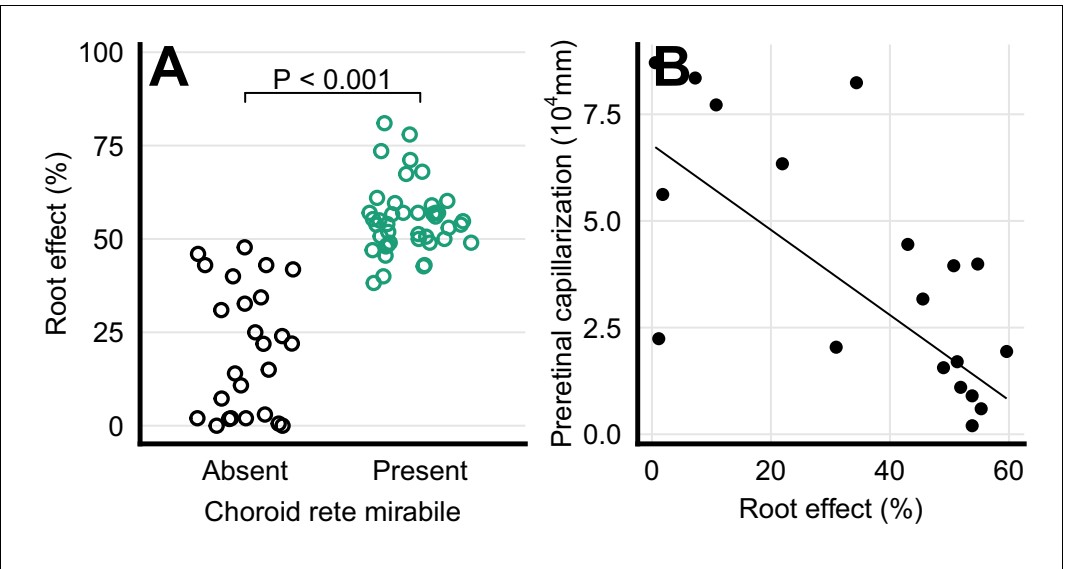

**Figure 5.** Haemoglobin function and retinal capillarization. (**A**) Frequency distribution of Root effect magnitude in ray-finned fishes with and without a choroid rete mirabile. The effect of the presence of the choroid rete mirabile was tested by phylogenetic analysis of variance simulation (F = 122, p<0.001, n = 68). (**B**) Pre-retinal capillarization and Root effect magnitude in ray-finned fishes, showing a negative correlation as tested by phylogenetic general least squares (t = -4.62, p<0.001, n = 19). Each dot represents mean values for each species. Pre-retinal capillarization is the volume of capillaries on the inner side in $mm^3$ of the retina per retinal surface area in $mm^2$. Root effect is the per cent haemoglobin desaturation at pH 5.5 compared to pH 8.5 in air-equilibrated buffers.

follow the pattern of retinal thickness correlating positively to oxygen supply capacity and further, where one type of oxygen delivery mechanism has been diminished it has been replaced by other types of oxygen delivery.

## Discussion

### The evolutionary interplay between retinal oxygen supply and eye morphology

Our analysis reveals a tight, mutually dependent relationship between retinal oxygen supply mechanisms and retinal morphology in ray-finned fishes and mammals. These relationships are revealed both in the statistical correlations, in the temporal overlap within macroevolutionary reconstructions, and in the cases of 'natural knock-out' within *Astyanax* and Channichthyidae. These three separate lines of data independently support the hypothesis that the origin of additional oxygen supply mechanisms was associated with the evolutionary increases in retinal thickness.

The evolutionary reconstruction indicates that the last common ancestor of bony fishes only supplied the retina with oxygen from the choriocapillaris. This finding contrasts previous investigations suggesting an earlier origin of the choroid rete mirabile in a shared ancestor of bony fishes (*Bailes et al., 2006*; *Yu et al., 2009*). Further, our reconstruction revealed that this ancient bony fish likely possessed a thin retina, typical for extant species with no additional oxygen supply mechanisms, as well as species with reduced eye size (*Figure 2B*). Given that the positive relationship between eye size and retinal thickness held true for early vertebrates, our prediction of a thin retina is consistent with the fossil evidence where the oldest known bony fishes, for example †*Guiyu oneiros* (*Zhu et al., 2009*) and the oldest lobe-finned fishes (*MacIver et al., 2017*) possessed small eyes. Therefore, we propose that the advanced retinal oxygenation patterns in extant bony fishes derive from a very basic physiological phenotype in their most recent common ancestor.

The trajectories of retinal evolution from the most recent common ancestor of bony fish to teleosts with oxygen secretion show that the origin of the choroid rete mirabile ~200 MYA coincided with significantly increased retinal thickness (e.g., *Figure 6A*). This finding strongly suggests that

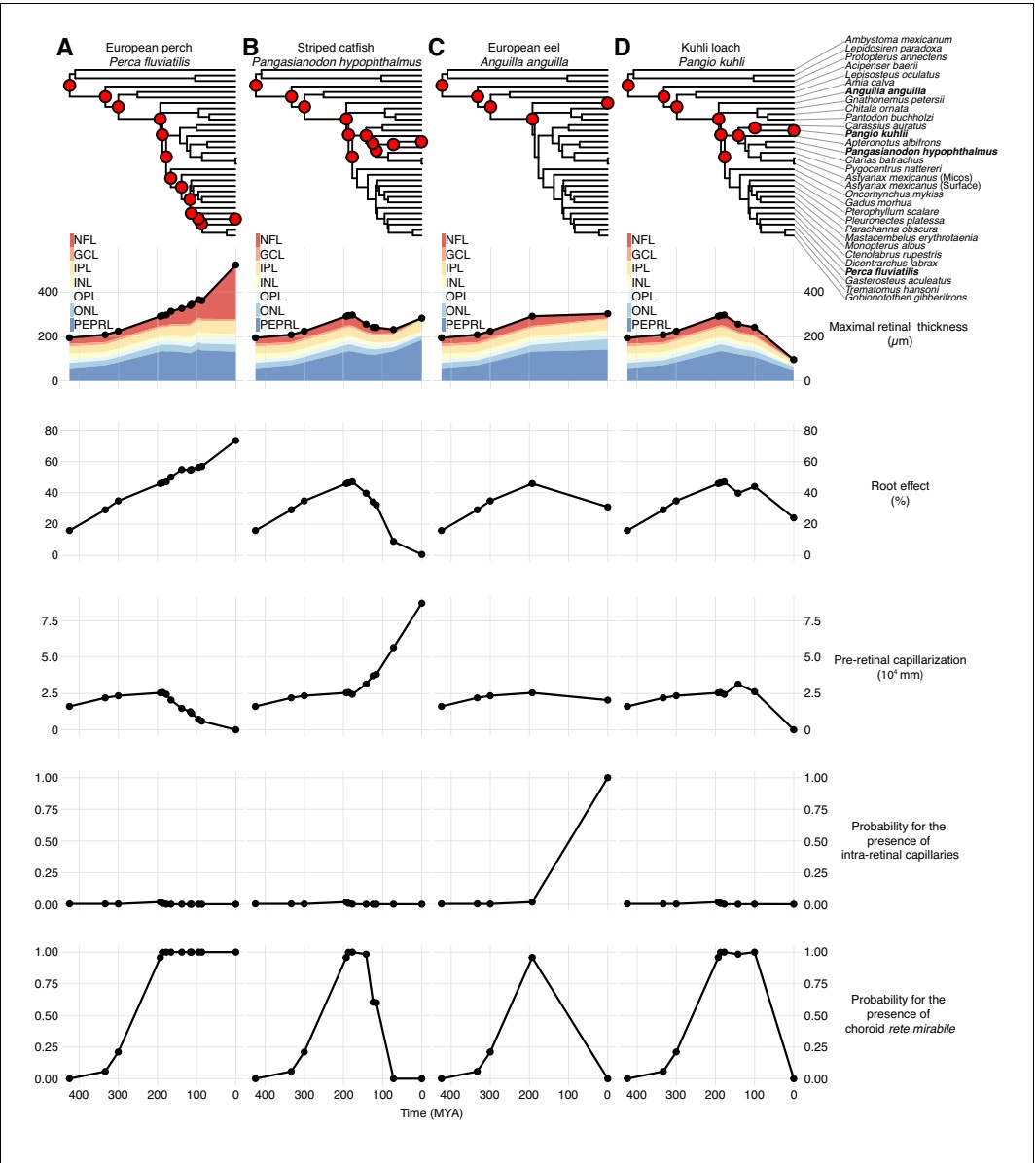

**Figure 6.** Evolutionary trajectories of retinal oxygen supply and morphology. Each column displays the evolution of physiological and anatomical parameters in the lines of descent connecting the most recent common ancestor of bony fishes to either European perch, *Perca fluviatilis* (A), striped catfish, *Pangasianodon hypophthalmus* (B), European eel, *Anguilla anguilla* (C), or kuhli loach, *Pangio kuhli* (D). The right-most symbols in each column are measured values and points to the left are reconstructed values of in all internal nodes in the phylogeny connecting to the most recent common ancestor of bony fishes (see top phylogeny, where the four species are marked in bold). Maximal retinal thickness is plotted in black where stacked shaded areas below represent measured and reconstructed thickness of the individual retinal layers. Pre-retinal capillarization is the volume of capillaries on the inner side in $mm^3$ of the retina per retinal surface area in $mm^2$. Root effect is the per cent haemoglobin desaturation at pH 5.5 compared to pH 8.5 in air-equilibrated buffers. Maximum total retinal thickness, thickness of individual retinal layers, and pre-retinal capillarization were reconstructed using maximum likelihood, and the presence of intra-retinal capillaries or the choroid rete mirabile was reconstructed using stochastic character mapping. Retinal layer abbreviations: PEPRL, pigment epithelium and photo receptor layer; ONL, outer nuclear layer; OPL, outer plexiform layer; INL, inner nuclear layer; IPL, inner plexiform layer; GCL, ganglion cell layer; NFL, nerve fiber layer. Trajectories to all extant species in the data set are deposited on GitHub (https://github.com/christiandamsgaard/Retinaevolution).

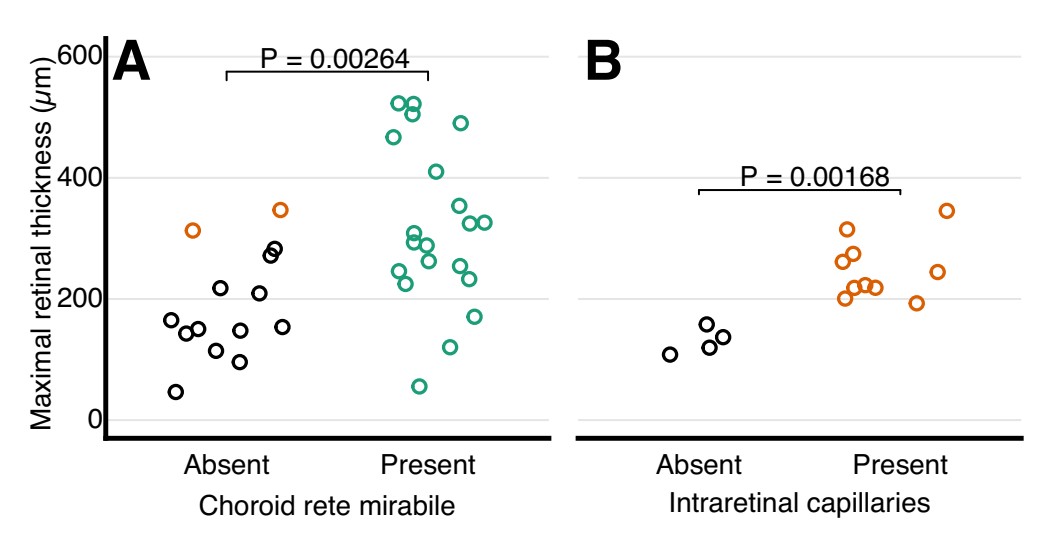

**Figure 7.** Relationship between retinal oxygen supply and morphology. Frequency distribution of retinal thickness in ray-finned fishes with and without a choroid rete mirabile (**A**), and in mammals with and without intra-retinal capillaries (**B**). In (**A**), orange symbols indicate species with intra-retinal capillarization. Each dot represents mean values for each species. The effect of the choroid rete mirabile or intra-retinal capillaries on retinal thickness were assessed by phylogenetic analysis of variance simulation. Root effect is the per cent haemoglobin desaturation at pH 5.5 compared to pH 8.5 in air-equilibrated buffers.

oxygen secretion conferred an adaptive advantage to teleosts by improving retinal oxygenation through a steeper diffusion gradient and hence permitted the morphological evolution of the retina. Further, our confirmation of the second hypothesis of a negative correlation between Root effect magnitude and pre-retinal capillarization suggests that oxygen secretion conferred an additional advantage to fish vision by reducing the adverse effects of light scattering from red blood cells in the visual field (*Country, 2017*; *Yu and Cringle, 2001*).

Our data set also revealed multiple independent losses of the oxygen secretion system within the teleosts and associated regressions in retinal morphology (*Figures 1* and *3*). The causality underlying these secondary losses in oxygen secretion and the declines in retinal thickness is challenging to resolve. However, our demonstration of reduced Root effect magnitude and regressed choroid rete mirabile morphology in cave-dwelling ecotypes of Mexican tetras proposes that the recurrent secondary losses of oxygen secretion across teleosts may have resulted from relaxed selection on vision. The rapid loss of Root effect may also be an indication of costs associated with the maintenance of enhanced Root effect haemoglobins. Such costs have been suggested previously and include exaggerated coupling between $CO_2$ and $O_2$ transport, increased oxidative damage, and a reduction in blood's oxygen-carrying capacity under acidification of the blood (e.g., under anaerobic exercise, hypoxia, hypercapnia) (*Pelster and Weber, 1991*; *Damsgaard et al., 2019b*; *Wilhelm Filho, 2007*). However, the loss of a choroid rete mirabile did not lead to a complete loss of the Root effect in species that utilise a rete mirabile in the swim bladder for buoyancy regulation, as these species tend to have a reduced, but not lost, Root effect in their blood (*Berenbrink et al., 2005*). Whatever the dynamics of the Root effect loss, the positive correlation between retinal thickness and the magnitude of the Root effect in Mexican cave fishes provide further support for an intrinsic link between retinal morphology and oxygen flux mechanisms within vertebrates.

Interestingly, our analysis identified some species descending from ancestors that had lost the choroid rete mirabile but retained a thick retina. These species provided valuable insight into alternative mechanisms for retinal oxygenation that are independent of oxygen secretion. Here, we identified multiple independent increases in pre-retinal capillarization downstream to separate secondary losses of oxygen secretion, including, but not limited to, the striped catfish and the Channichthyidae. This vascular pattern also originated within the lobe-finned fishes that never evolved oxygen secretion, such as within the amphibians, reptiles, and birds (*Country, 2017*; *Yu et al., 2009*;

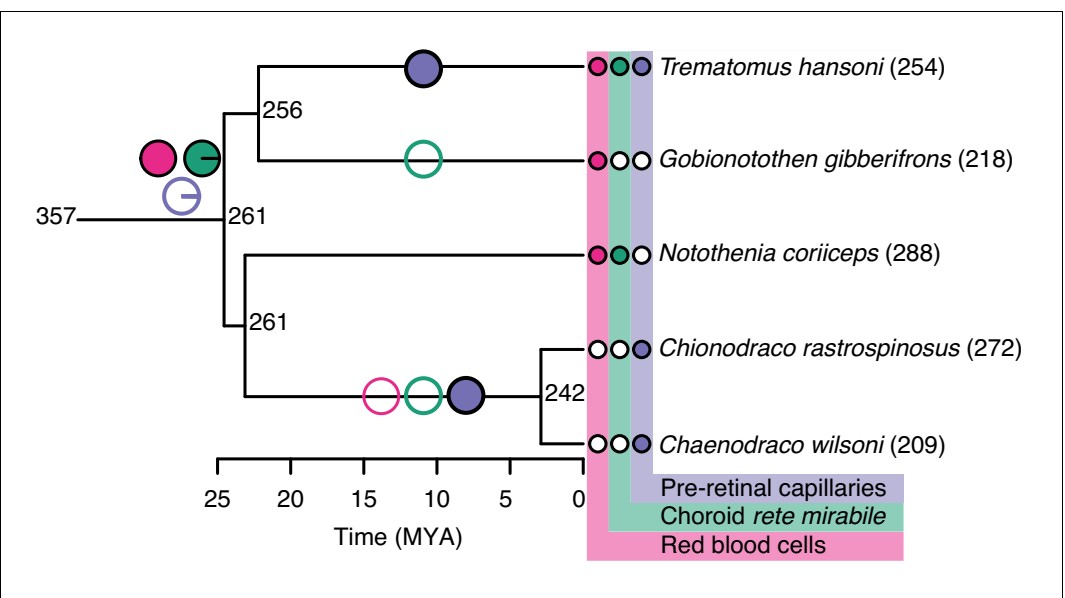

**Figure 8.** Evolution of retinal oxygen supply and morphology in Notothenioids, including the haemoglobin-less Antarctic icefishes (*C. rastrospinosus* and *C. wilsoni*). Measured and reconstructed values of maximal retinal thickness plotted on the Notothenioid phylogeny. Ancestral state values for retinal thickness were estimated by maximum likelihood. The observed absence (open symbols) and presence (filled symbols) of red blood cells (pink), choroid rete mirabile (green), and pre-retinal capillaries (purple) are indicated at the tips, and inferred origins and losses of these traits are marked on internal branches by filled and open symbols, respectively. Pie charts at the most recent common ancestor of notothenioids indicate the Bayesian posterior probability of these traits being present.

*Meyer, 1977*), providing the same general mechanism for improved retinal oxygenation, which was also associated with retinal thicknesses exceeding those of many other tetrapods.

Intra-retinal capillaries represent an alternative mechanism to improve oxygen supply in the absence of oxygen secretion. The two independent origins of intra-retinal capillaries within teleosts both overlapped with the losses of oxygen secretion, but a maintenance of retinal thickness, which illustrates the efficacy of intra-retinal capillaries in oxygenating the retina. This retinal capillarization is functionally similar to the solution that evolved independently in mammals. Interestingly, the independent origins of holangiotic capillarization in mammals seem to coincide with the origin of endothermy in a late-Mesozoic dinosaur group, whereby mammals altered from a nocturnal to diurnal activity pattern that increased the reliance on vision and a thicker retina (*Gerkema et al., 2013*). Our reconstruction also revealed five secondary losses of intra-retinal capillaries in the eutherian mammal groups, which include species that have evolved non-visual primary senses (e.g., echolocation in bats) with retinal thicknesses well below those of species with intra-retinal capillarization. These observations all support the second hypothesis of a tight connection in the evolution of enhanced retinal oxygen delivery mechanisms and improved retinal thickness.

## Retinal oxygen supply and the evolution of visual performance

The six-fold interspecific differences in retinal thickness raise the question of the adaptive significance of retinal thickness to visual performance. Several studies have shown that retinal thickness increases with the density of photoreceptors and retinal ganglion cells, where the latter relates directly to spatial resolution (*Walls, 1937*; *Querubin et al., 2009*; *Potier et al., 2017*). This relationship is particularly pronounced in the retinal region surrounding the fovea or area centralis, where visual cells are not only densely packed but also frequently arranged in several layers, one above the other (*Querubin et al., 2009*; *Potier et al., 2017*; *Jeffery and Williams, 1994*). The only exception to this is the central fovea, where the retinal ganglion cells (and sometimes other inner retinal layers) are partially or fully displaced to create a depression on the retinal surface (known as the fovea), below which the retina is much thinner. Nonetheless, a thicker retina with a higher density of visual

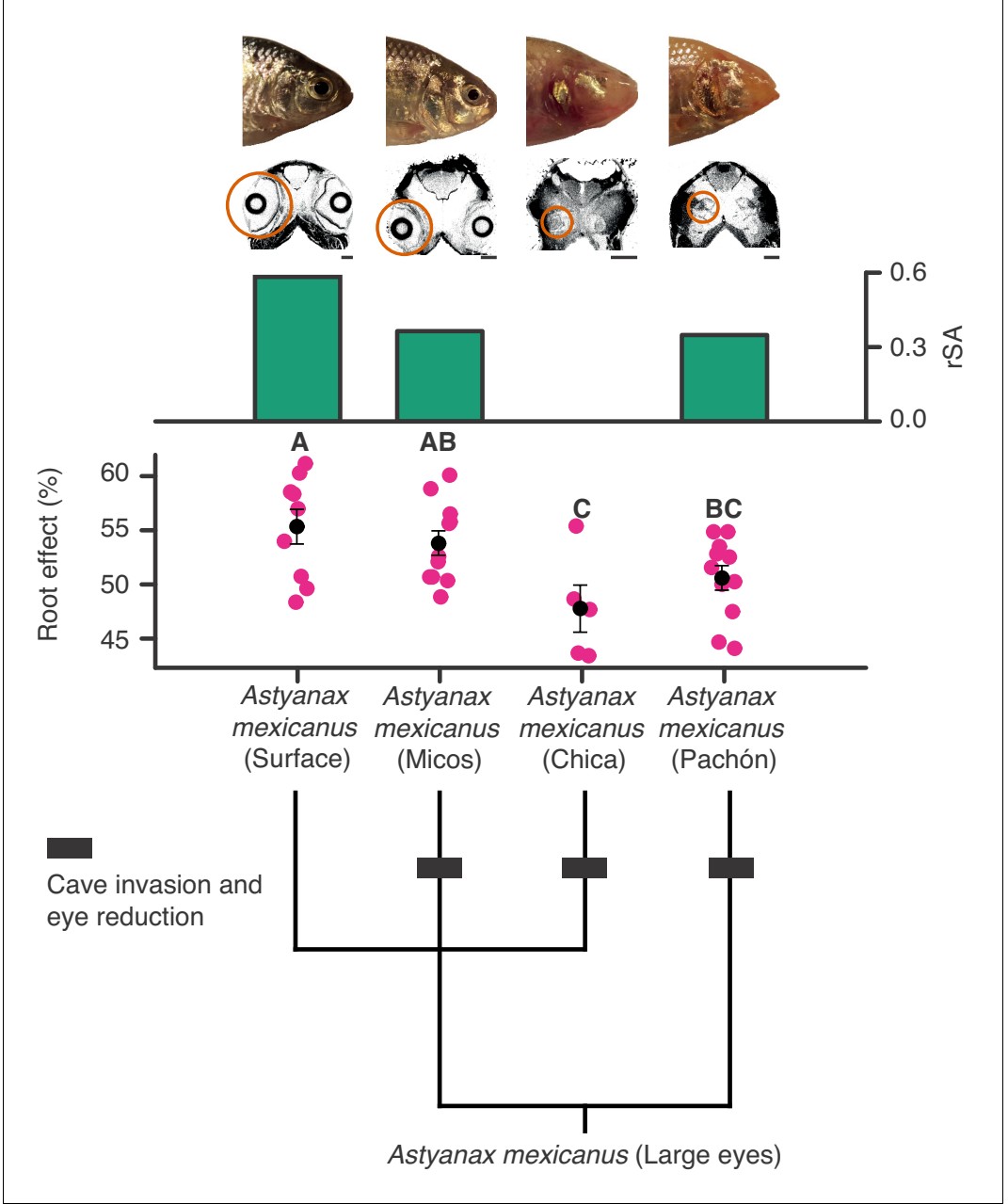

**Figure 9.** Regressive evolution of oxygen secretion in troglobitic Mexican tetras. Representative photographs and computed tomography scans of a surface and three cave forms of *Astyanax mexicanus*: Surface (large eyes), Micos (invaded caves about 10–20,000 years ago, variably reduced eyes), Chica (invaded caves around 10–20,000 years ago, highly reduced eyes), and Pachón (invaded caves at earliest around 3 Ma but most likely during the last glaciation, highly reduced eyes). Orange circles on computed tomography scans mark the actual size and location of eyes. Green bars show residuals of $\log_{10}$(choroid rete mirabile endothelium surface area [mm]) (rSA) on $\log_{10}$(body mass [g]). There was no choroid rete mirabile in *A. mexicanus* (Chica). Root effects magnitudes are marked as pink dots, and black dots and bars are means and standard error of mean. There were significant differences in mean Root effect between ecotypes as determined by one-way ANOVA ($F_{3,\ 32}$ = 4.617, p=0.009) with a Student-Newman-Keuls posthoc test and indicated by letters above bars (n = 9, 11, 11, and five for Surface, Micos, Pachón and Chica, respectively). Lower panel shows the phylogenetic relationships between different forms of *A. mexicanus*.

cells arranged in layers (as found in the vicinity of a fovea or area centralis) confers a number of

performance advantages: Firstly, the ability to resolve spatial details in a well-focused retinal image is directly proportional to the density of ganglion cells (*i.e.*, to the fineness of the sampling matrix; *Rodieck, 1998*). Secondly, layers of ganglion cells (possibly with parallel sampling matrices) have the potential to analyze different types of visual information in parallel pathways (e.g. the parallel 'ON' and 'OFF' ganglion cell pathways; *Rodieck, 1998*). Thirdly, a thicker photoreceptor layer allows the possibility of longer rod and cone outer segments for greater light absorption and enhanced sensitivity (*Land and Nilsson, 2012*; *Cronin, 2014*). Indeed, in many species of deep-sea fishes (*Warrant and Locket, 2004*; *Locket, 1977*), and in one remarkable nocturnal bird (the oilbird, *Steatornis caripensis*; *Martin et al., 2004*), the retina has thickened substantially to allow several layers of rods, hence significantly improving photon catch in the light-impoverished environments where these animals live.

## Summary

A thicker retina obviously confers multiple performance advantages to the vision of animals, but simultaneously introduces an inherent trade-off between visual performance and retinal oxygen delivery. Here, we provide evidence for the convergent evolution of improved retinal morphology from a primitive retinal phenotype that was invariably associated with the emergence of improved oxygen supply mechanisms. We show that these additional mechanisms have been repeatedly gained and lost during vertebrate evolution and are in all cases associated with parallel changes in retinal morphology and possibly visual performance. Based on these data, we propose that retinal oxygen diffusion constrained the evolution of improved vision in ancestral vertebrates; a constraint that was repeatedly relaxed by various combinations of vascular and haemoglobin adaptations permitting the adaptive evolution of the vertebrate eye.

# Materials and methods

## Study design

We used an integrative and comparative analysis of retinal morphology and respiratory phenotyping. First, measurements of the functional anatomy of the vertebrate eye were made in vivo with high-frequency ultrasound (*Videos 1–2*). These were expanded with stereological analyses of histological sections, high-resolution computed tomography and magnetic resonance imaging of whole eyes to quantify extra-retinal capillarization and thicknesses of all retinal layers and were combined with measurements of haemoglobin functional properties (*Figure 3—figure supplement 1*). We analyzed these data in a phylogenetic comparative framework to test to what extent independent origins of additional retinal oxygen supply were consistently associated with improved retinal morphology. When compatible, we further included published literature values for retinal thickness and Root effect magnitude and the absence and presence of a choroid rete mirabile and intra-retinal capillaries to increase the power in the ancestral state reconstructions (*Supplementary file 1*).

## Animal procurement and housing

Eighty-seven species of fishes were included in this study, and were chosen to include representatives from as many clades as possible that were known to contain species with and without distinct types of retinal oxygen supplies. Of the initial 87 species, 31 species were examined by in vivo ultrasound imaging. These specimens were purchased from local aquaculture facilities, research institutions, or aquarium stores, and kept in large tanks coupled with a recirculation system at Zoophysiology, Aarhus University, Denmark. Animals were fed daily with commercially dry pellets until analysis and held under a 12:12 hr light cycle. Animals of the three ecotypes of *Astyanax mexicanus* (Surface, Micos, and Pachón) were offspring of animals caught in their natural caves by H.W. Notothenioid specimens were captured and sampled by TD and HWD along the West Antarctic Peninsula during a field campaign in 2016 supported by the United States Antarctic Program (USAP).

Several specimens with the following museum numbers were loaned from Natural History Museum of Denmark to examine retinal capillarity phenotype using whole animal magnetic resonance imaging in *Latimeria chalumnae* (ZMUC P1112, Conservation du Coelacanthe number 23) and *Dissostichus eleginoides* (ZMUC P63215), and using micro-CT following by histological sectioning of the eye of *Neoceratodus fosteri* (ZMUC P1127) (*Figure 3—figure supplement 2*).

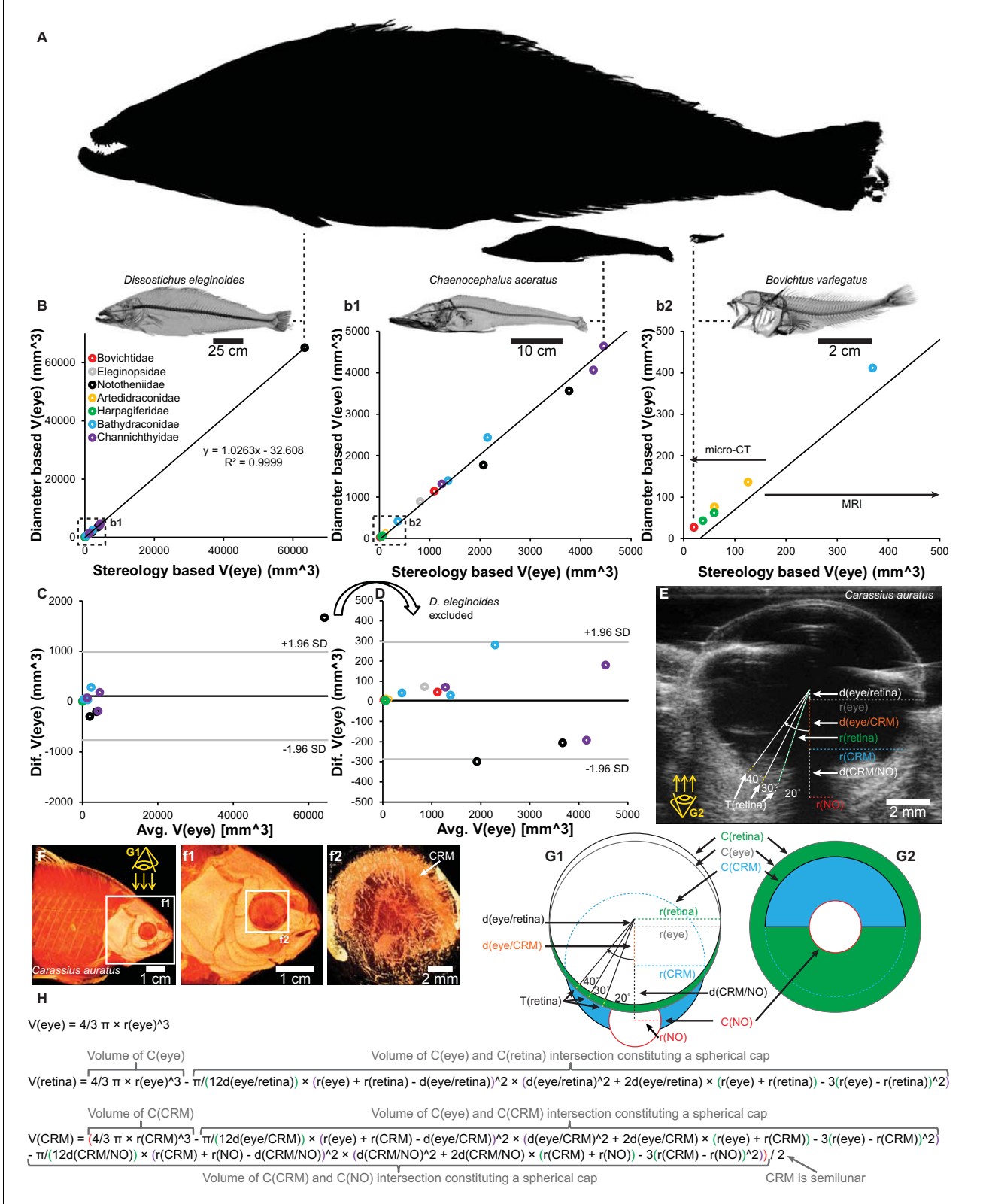

**Figure 10.** Validation of eye volume [*V*(eye)] measurement from eye radius and schematic of quantitative anatomical measurements. Estimations of *V*(eye) in 16 species of notothenioids spanning 4 orders of magnitude in body size (A, relative sizes of largest, a medium sized and smallest species) showed a proportional relationship between *V*(eye) based on eye radius and stereological measurements from three-dimensional magnetic resonance imaging (MRI) and micro-CT imaging (B, correlation plot with magnifications (b1 and b2). *C* and *D*, Bland-Altman plot containing all specimens (C) and

*Figure 10 continued on next page*

*Figure 10 continued*

with the markedly larger *Dissostichus eleginoides* excluded (D)). Retinal thickness, T(retina), was measured at 20°, 30° and 40° to the optic nerve at both sides (only one side shown in figure) on ultrasound (E) and micro-CT (F, volume reconstruction to show shape of choroid rete mirabile) datasets and similarly on histological slides. Eye volume, retina volume, *V*(retina) and choroid rete mirabile volume, *V*(CRM), were calculated based on geometrical assumptions (G1, G2, and H. See Materials and methods for written description) and the measurement of the radii, r(x), of the circle describing the eye, *C*(eye), and the imaginary circles constituted by the inner side of the retina, *C*(retina), the choroid rete mirabile, *C*(CRM), the optic nerve, *C*(NO) and the displacement, *d*(x), of the center of these circles.

To test for the effect of spheriality on eye volume estimations, we used whole animal imaging data sets of nototenioid specimens with the following museum numbers that were scanned using magnetic resonance imaging in *Cottoperca gobio* (ZMUC P639); *Eleginops maclovinus* (ZMUC CN2); *Notothenia angustata* (ZMUC 7746); *Notothenia coriiceps* (ZMUC 9086); *Gymnodraco acuticeps* (ZMUC P6346); *Parachaenichthys georgianus* (ZMUC P632), *Parachaenichthys charcoti* (ZMUC 6); *Champsocephalus gunnari* (ZMUC P63195); *Chaenocephalus aceratus* (ZMUC P63200); *Chionodraco rastrospinosus* (ZMUC P63202)], and using micro-computed tomography imaging (micro-CT) in *Bovichtus variegatus* (ZMUC CN1), *Artedidraco skottsbergi* (ZMUC 8374), *Pogonophryne immaculate* (ZMUC 7755), *Harpagifer antarcticus* (ZMUC P63273), and *Harpagifer bispinis* (ZMUC P6310).

## Micro-ultrasound imaging

High-frequency micro-ultrasound was applied to visualise retinal blood flow in vivo. Fish were lightly anaesthetised until unresponsive to tactile stimulation using benzocaine (ethyl-4-aminobenzoate) at species-dependent concentrations (*Supplementary file 2*) and immobilised in a custom-made harness underwater and under the same light regime. Ultrasound imaging was performed using a VisualSonics Vevo 2100 system with either a 21 MHz transducer (MS250) for very large species, a 40 MHz transducer (MS550d) for medium-sized species, or a 48 MHz transducer (MS700) for smaller species (*Supplementary file 2*), with the transducer placed orthogonally to the eye and parallel to the longitudinal axis of the fish. Both eyes were imaged 1 cm underwater using brightness mode to acquire high-resolution transversal videos and blood-motion-enhanced images by quadratic averaging (described below), and colour Doppler mode to acquire colour-coded videos of directional blood movement. All individuals within a species had similar blood flow patterns. Ultrasonographic videos from all species can be found in *Videos 1–2*. The transducer was initially translated across the entire eye surface to scan for optimal positioning of 2D ultrasound section for quantitative measurements.

Ultrasound imaging was used to quantify maximal retinal thickness [*T*(retina)] (as described below in *Stereological analysis* section) and the volumes of the eye [*V*(eye)], the retina [*V*(retina)], and the choroid rete mirabile [*V*(CRM)]. Maximal retinal thickness was used in the subsequent data analysis, while *V*(eye), *V*(retina), and *V*(CRM) were used to assess intraspecific variation in eye morphology (*Supplementary file 3*). For *V*(eye), a spherical eye shape was assumed, and volume was calculated from the radius [*r*(eye)] of the eye:

$$V(eye) = \frac{4}{3}\pi r(eye)^3 \tag{1}$$

To determine how slight deviations from sphericity biased the results, 16 notothenioid species previously harvested as intact museum specimens were subjected to detailed analysis of the relationship between geometrically calculated eye volume from average eye radius (average of horizontal and vertical radius of the eye) and eye volume calculated from stereological measurements on 3D datasets acquired with MRI and micro-CT (see description of scanning parameters below). This showed that eye volume estimates based on eye radius were proportional (*Figure 10*) and not significantly different (p=0.33, paired t-test) to *V*(eye) estimates obtained by three-dimensional stereological analysis, thus allowing valid inclusion of species where only eye radius was available.

Retinal volume estimation was performed by subtracting the volume of intersection between the imaginary sphere constituted by the inner side of the retina [with the radius *r*(retina)] and the eye sphere displaced by the maximum thickness of the retina [*d*(eye/retina)] from the total eye volume (*Figure 10*):

$$V(\text{retina}) = \tfrac{4}{3}\pi r(\text{eye})^3 - \tfrac{\pi}{12d(\text{eye/retina})} \times (r(\text{eye}) + r(\text{retina}) - d(\text{eye/retina}))^2 \times$$
$$\left( d(\text{eye/retina})^2 + 2d(\text{eye/retina})(r(\text{eye}) + r(\text{retina})) - 3(r(\text{eye}) - r(\text{retina}))^2 \right) \tag{2}$$

Choroid rete mirabile volume estimation was performed by assuming a semilunar shape of the choroid rete mirabile and subtracting the volume of intersection between the imaginary sphere constituted by the choroid rete mirabile [with the radius $r$(CRM)] and the eye sphere displaced by $d$(eye/CRM) and the volume of intersection between the imaginary sphere constituted by the choroid rete mirabile and the imaginary sphere constituted by the optic nerve penetrating though the choroid rete mirabile [with the radius $r$(NO)] displaced by $d$(CRM/NO) from the total imaginary CRM sphere volume (**Figure 10**):

$$V(\text{CRM}) = (\tfrac{4}{3}\pi r(\text{CRM})^3 - \tfrac{\pi}{12d(\text{eye/CRM})} \times (r(\text{eye}) + r(\text{CRM}) - d(\text{eye/CRM}))^2 \times$$
$$(d(\text{eye/CRM})^2 + 2d(\text{eye/CRM})(r(\text{eye}) + r(\text{CRM})) - 3(r(\text{eye}) - r(\text{CRM}))^2) -$$
$$\tfrac{\pi}{12d(\text{CRM/NO})} \times (r(\text{CRM}) + r(\text{NO}) - d(\text{CRM/NO}))^2 \times (d(\text{CRM/NO})^2 + 2d(\text{CRM/}$$
$$\text{NO})(r(\text{CRM}) + r(\text{NO})) - 3(r(\text{CRM}) - r(\text{NO}))^2))/2 \tag{3}$$

Both $T$(retina), $V$(eye), $V$(retina), and $V$(CRM), showed low intraspecific variation (**Supplementary file 3**), and consequently, only one replicate per species was used in the stereological analysis of micro-CT and histology. Additionally, 3D ultrasound was used to model the blood flow architecture in five representative species of different retina supply types (**Video 2**). Here a VisualSonics Vevo 2100 system with a 48 MHz transducer (MS700) was moved using motor-assisted stepwise 2D scanning in 20 µm steps to scan the entire eye in the dorsal-to-ventral direction, acquiring 1000 frames/step. Each sectional dataset was post-processed to calculate the quadratic averaged intensity ($D$) from the arithmetic averaged intensity ($\hat{I}$) from the intensity ($I$) at each pixel coordinate ($x$, $y$) over time ($t$) in the total number of frames ($N$):

$$D(x,y) = \left[ \frac{1}{N} \sum_{t=1}^{N} \left( I(x,y,t) - I(x,y,t) \right)^2 \right]^{\frac{1}{2}} \tag{4}$$

This was reconstructed into 3D datasets with a $30 \times 30 \times 20$ µm$^3$ spatial resolution in which blood speckles that are much more dynamic than tissue speckles were greatly enhanced, thereby creating a functional angiography of the eye without the use of contrast agent (**Tan et al., 2015**).

## Blood- and eye sampling

After micro-ultrasound imaging, animals were euthanised with a benzocaine overdose in aquaria water. Blood was withdrawn from the caudal vein using a heparinised (5,000 IU ml$^{-1}$) syringe in larger specimens and using heparinised hematocrit tubes from the exposed heart ventricle in smaller specimens. Fishes were then decapitated, and the whole heads fixed by immersion in buffered 4% formaldehyde for 10 min. Both eyes were then removed, weighed and immersed in buffered 4% formaldehyde at 5˚C until histological preparation.

## Root effect

Immediately after sampling, blood was centrifuged (3 min, 2,900 $g$) to remove plasma and leukocytes. The red blood cell pellet was washed three times in saline and then stored and lysed at −80˚C. Red blood cell solutions were thawed on ice, centrifuged (1 min, 12,000 $g$), and mixed to a final heme concentration of ~10 µmol l$^{-1}$ in either 50 mmol l$^{-1}$ citrate-HCl (pH 5.5) or Tris-HCl buffers (pH 8.5) containing 100 mmol l$^{-1}$ KCl. Root effect magnitude was calculated as the per cent desaturation at pH 5.5 compared to pH 8.5, by recording the absorbance spectrum between 480 and 700 nm at 0.2 nm intervals, where fractions of oxyHb, deoxy-Hb and metHb were calculated by spectral deconvolution using species-specific wavelength spectra of oxyHb, deoxy-Hb and metHb (**Jensen, 2007**). This method shows the same Root effect magnitude as in whole blood (**Berenbrink et al., 2011**).

## Micro-computed tomography and magnetic resonance imaging

Micro-CT was used to acquire high-resolution three-dimensional information of ocular anatomy. In order to produce soft-tissue contrast in the micro-CT images, the fixed eye samples were soaked for 1–3 days (depending on specimen size) in phosphate-buffered saline to remove residual formaldehyde, and then immersed in isosmotic Lugol's solution (16.6 g $l^{-1}$ KI and 8.3 g $l^{-1}$ $I_2$ in distilled water) for 2–14 days (depending on specimen size) to ensure adequate iodine staining of the samples. Micro-CT imaging was performed using a Scanco Medical µCT 35 scanner (Scanco Medical AG, Brüttisellen, Switzerland) in high-resolution mode (1000 projections/180°) with an isotropic voxel size of 3.5 µm, 6 µm, 10 µm, or 15 µm (depending on specimen size; *Supplementary file 2*), an X-ray tube voltage of 70 kVp, an X-ray tube current of 114 µA, and an integration time of 800 ms. Arterial perfusion via the ventral aorta using a custom made, $BaSO_4$-containing, CT contrast agent (*Rasmussen et al., 2010*) was applied to model the vascular supply of the eye in goldfish. This specimen was imaged on an Xradia Zeiss VersaXRM-520 system to acquire both a low-resolution scan of the entire head region at 120 kVp with 1200 projection/180° and an isotropic voxel size of 36.5 µm and a high-resolution scan of the eye region at 100 kVp with 1200 projection/180° and an isotropic voxel size of 4.6 µm. For validation of eye sphericity five smaller notothenioid whole-body specimens were micro-CT scanned on a clinical Scanco Medical XtremeCT system (Scanco Medical AG, Brüttisellen, Switzerland) with 1500 projections/180° and an isotropic voxel size of 41 µm, an X-ray tube voltage of 59.4 kVp, an X-ray type current of 119 µA and an integration time of 132 ms.

For larger intact notothenioid species and the intact coelacanth specimen, MRI was performed on a clinical 3 T Siemens Magnetom Skyra system (Siemens Medical Solutions, Forchheim, Germany). For each scan, the fish were positioned on one side, and multiple surface RF coils were applied to cover the specimen. For both notothenioids and the coelacanth a 3D gradient echo DIXON sequence was first acquired with the following parameters: Repetition time (TR) = 5 ms, echo time (TE) = 1.23 ms, excitation flip angle of 10°, three averages and an isotropic image resolution of 1.42 mm. The eye of the coelacanth, still in place in the skull, was subsequently imaged applying a small surface RF coil with two sequences: first a T1 weighted gradient echo sequence with the following parameters: TR = 40 ms, TE = 8.53 ms, excitation flip angle = 35°, and an isotropic image resolution of 400 µm; secondly a T2 weighted spin-echo sequence with the following parameters: TR = 1.03 s, TE = 136 ms, four averages, and an isotropic image resolution of 350 µm. Then the left eye of the coelacanth was dissected free and was MRI scanned at a higher field strength in a pre-clinical 9.4 T Agilent system (Agilent Technologies, Oxford, United Kingdom) using a Rapid 72 surface RF coild and applying two sequences: first a T1 weighted gradient-echo sequence with the following parameters: TR = 11 ms, TE = 5.87 ms, excitation flip angle = 30°, eight averages, and an isotropic image resolution of 121 µm; secondly a T2 weighted turbo spin echo sequence with the following parameters: TR = 1 s, TE = 41.7 ms, echo train length = 8, four averages, and an isotropic image resolution of 242 µm. ImageJ (Wayne Rasband, National Institutes of Health, USA) version 1.51 hr was used for micro-CT, MRI and ultrasound data analysis and Amira 5.6 (FEI, Visualization Sciences Group) was used for anatomical model building.

## Histology

Formaldehyde-fixed eyes were dehydrated in an ethanol series (70%, 96%, 99.9%), embedded in paraffin and sectioned at 5-µm-thick along the ventral-to-dorsal axis. The tiny eyes (*Lepidosiren paradoxa*, *Pangio kuhlii*, *Astyanax mexicanus*, *Gasterosteus aculeatus*) were fixed in situ and a portion of the head containing the eye, sectioned after decalcification in EDTA. All sections were stained with hematoxylin and eosin. Sections were imaged using bright-field microscopy with 20× (NA = 0.75) or 60× (NA = 1.35) oil objectives on an Olympus VS120 virtual slide scanning system.

## Stereological analysis

$V$(CRM), $V$(retina), and $V$(eye) were estimated from the micro-CT scans by choosing $\geq$12 equally spaced parallel sections spanning each eye and applying a systematic uniform point grid on each section plane. Volumes were calculated by counting the number of points intersecting each tissue using the Cavalieri estimator (*Mühlfeld et al., 2010*). Similarly, the volume of pre-retinal capillaries [$V$(PRC)] was found by estimating the volume densities using test points on histological sections and multiplying by $V$(retina) estimated from micro-CT scans. The criterion for the presence of pre-retinal

capillaries was set at vessels on the inner side of the retina with a diameter <3 times the length of the long semi-axis of the red blood cell spheroid. Presence of red blood cells in the retina was used as the criterion for the presence of intra-retinal capillaries. Types of intra-retinal capillaries in mammals were taken from the literature (*Chase, 1982*; *Leber, 1903*; *Samorajski et al., 1966*; *Moritz et al., 2013*; *Kolmer, 1927*; *Bellhorn, 1997*; *McMenamin, 2007*).

Choroid rete mirabile surface area [*SA*(CRM)] and retinal surface area [*SA*(retina)] were estimated by applying a systematic set of parallel test lines (with length *L*) to >5 and>3 histological sections, respectively. The number of intersections (*I*) with the choroid rete mirabile surface endothelium and the retina were counted, and *SA*(CRM) and *SA*(retina) were estimated as $2 \times I \times V$(CRM)$/L$ and $2 \times I \times V$(retina)$/L$, respectively (*Mühlfeld et al., 2010*), using *V*(CRM) and *V*(retina) estimated from micro-CT scans. The spherical shape of the eye makes it possible to fulfil the assumption of isotropic vessel surfaces.

Maximum retinal thickness was measured as the maximum perpendicular distance through the retina from either 20, 30, or 40 degrees on each side of the optic nerve using ultrasound. To obtain information on the thickness of the individual retinal layers, we used the histological section that included the optical nerve. Here, the *relative* thicknesses of the individual retinal layers were measured from the position on the retina, where retinal thickness was maximal (choosing between positions 20, 30, or 40 degrees on each side of the optic nerve and Bruch's membrane and the inner limiting membrane as retinal boundaries), and *absolute* retinal layer thickness was calculated from the absolute retinal thickness obtained from ultrasound and relative layer thickness obtained from histology. This approach was chosen to avoid any potential bias from retinal shrinkage. However, maximal retinal thickness obtained from ultrasound and histology did not differ (p=0.18, paired t-test), and hence maximal retinal thickness and retinal layer thicknesses were obtained from histological sections in the notothenioids.

Stereological analyses of micro-CT scans and histological sections were performed in Fiji (1.8.0) and newCAST (Visiopharm, Hørsholm, Denmark), respectively.

## Statistical analysis

A composite phylogeny was generated based on published, time-calibrated phylogenies, pruned to represent only species within the data set.

A simulation-based phylogenetic ANOVA was used to test for the effect of intra-retinal vessels or choroid rete mirabile on retinal thickness using a phylogenetic analysis of variance simulation, where a null distribution of F-statistics was simulated by allowing traits to evolve by Brownian motion across the tree for 50,000 generations, and calculated the probability of the observed F-statistic lying within the null distribution of F-statistics (α <0.05) (*Garland et al., 1993*).

Generalised least-squares fit by maximum likelihood were used to test for correlations between continuous characters assuming an error structure that follows either an Ornstein-Uhlenbeck or Brownian motion model for character evolution (Phylogenetic general least squares, PGLS), and the best fit was chosen based on Akaike's weight.

Eye mass was body mass corrected by computing the residuals from a PGLS regression of species-mean $\log_{10}$(eye mass [g]) plotted against species-mean $\log_{10}$(body mass [g]). The choroid rete mirabile endothelium surface area was corrected for body mass in the same way.

The evolutionary histories of continuous characters were reconstructed using maximum likelihood using a Brownian motion model for character evolution. To reconstruct the evolutionary history of the choroid rete mirabile as well as intra- and pre-retinal capillaries, we summarised information from 10,000 simulations of stochastic character mapping with transition rate matrix with equal rates, and an equal root node prior distribution (*Bollback, 2006*). Branches on which transitions most likely occurred were identified by first mapping the Bayesian posterior probabilities onto the phylogeny and then identifying branches where the probabilities changed between < 0.5 and > 0.5 along the branch. The reconstruction of intra-retinal capillarization across all species in the data set suggested anangiotic capillarization of the retina in the most recent common ancestor of mammals. Thus, the reconstruction of intra-retinal capillarization within mammals on *Figure 4* was simulated using 1:0:0 root node prior distribution for anangiotic-, merangiotic-, and holangiotic capillarization for mammals.

Differences in Root effect magnitude between ecotypes of *Astyanax mexicanus* were determined by one-way ANOVA, and pairwise differences between groups were determined by a Student-

Newman-Keuls posthoc test ($\alpha$ < 0.05). Analyses were made R v. 3.5.0 using the **phytools**, **ape**, and **phangorm**, and **geiger** packages (*Revell, 2012*; *Paradis et al., 2004*; *Schliep, 2011*; *Harmon et al., 2008*).

The full raw data set, computer code, and evolutionary trajectories to all species in *Figure 6* are deposited on GitHub (https://github.com/christiandamsgaard/Retinaevolution; *Damsgaard et al., 2019a*; copy archived at https://github.com/elifesciences-publications/Retinaevolution).

## Ethics

The ultrasonography was performed with permission from the Danish Inspectorate for Animal Experimentation within the Danish Ministry of Food, Agriculture and Fisheries, Danish Veterinary and Food Administration (Permit number 2016-15-0201-00835). The Central Denmark Region Committees on Health Research Ethics approved the use of histological sections from human autopsies, project ID M-2012-299-12, file 48447. The University of Oregon Institutional Animal Care and Use Committee approved the capture and sampling of notothenioid specimens (Animal Welfare Assurance Number A-3009-01, IACUC protocol 12-02RA).

## Acknowledgements

We thank Aage Alstrup, The Natural History Museum of Denmark, Kattegatcentret, Nordsøens Oceanarium, Rachael Heuer, Timothy Bister, Jacob Green, Marcus Krag and Josh DeWyse for helping with supply of animals; Mads Andersen, Malthe Hvas, Jakob Thyrring, Rasmus Buchanan, and Heidi Meldgaard for help with animal handling; Kristian Beedholm for help with MatLab scripts; Olav M Andersen for providing access to mouse histological sections; Toke Bek for loaning histological sections from humans and for constructive feedback; Elin Pedersen and Helene Andersen for laboratory assistance; Michael W Country and Chris M Wood for feedback on earlier versions of the manuscript; Hannah Wright, Oliver Kepp, and MBe 's advanced undergraduate students at the Institute of Animal Physiology, Humboldt University, Berlin, for help with Root effect measurements from seven fish species.

The Danish Ministry of Foreign Affairs (DANIDA) [DFC no. 12-014AU] (MBa), the Carlsberg Foundation (CD), the Velux Foundation (HL), the UK Biotechnology and Biological Sciences Research Council (BBSRC 26/S17991) (MBe), the Danish Council for Independent Research (to TW), the US National Science Foundation Office of Polar Programs [NSF/PLR OPP-1444167 to HWD, and NSF/PLR OPP-1543383 to HWD and TD] funded the project. The Villum Foundation supported the Centre for Stochastic Geometry and Advanced Bioimaging. The VELUX Foundation and the Karen Elise Jensens Foundation donated the micro-CT scanner, and the ultrasound scanner, respectively. This is contribution 400 from the Northeastern University Marine Science Center.

## Additional information

### Funding

| Funder | Grant reference number | Author |
| --- | --- | --- |
| Danida Fellowship Centre | DFC no. 12-014AU | Mark Bayley |
| Carlsbergfondet | CF16-0713 | Christian Damsgaard |
| Carlsbergfondet | CF17-0195 | Christian Damsgaard |
| Biotechnology and Biological Sciences Research Council | BBSRC 26/S17991 | Michael Berenbrink |
| Danish Council for Independent Research | | Tobias Wang |
| National Science Foundation | Office of Polar Programs NSF/PLR OPP-1444167 | H William Detrich III |
| National Science Foundation | Office of Polar Programs NSF/PLR OPP-1543383 | Thomas Desvignes H William Detrich III |
| Villum Fonden | | Jens R Nyengaard |

| Velux Fonden | Jesper S Thomsen<br>Annemarie Brüel<br>Henrik Lauridsen |
| Karen Elise Jensens Fond | Henrik Lauridsen |

The funders had no role in design, execution, and submission of this study.

## Author contributions

Christian Damsgaard, Conceptualization, Data curation, Formal analysis, Funding acquisition, Validation, Investigation, Visualization, Methodology, Writing;original draft, Writing;review and editing; Henrik Lauridsen, Conceptualization, Resources, Software, Formal analysis, Funding acquisition, Validation, Investigation, Visualization, Methodology, Writing;original draft, Writing;review and editing; Anette MD Funder, Resources, Investigation, Methodology; Jesper S Thomsen, Resources, Funding acquisition, Investigation, Methodology, Writing;original draft, Writing;review and editing; Thomas Desvignes, H William Detrich III, Resources, Funding acquisition, Investigation, Writing;original draft, Writing;review and editing; Dane A Crossley II, Resources, Investigation, Writing;original draft, Writing;review and editing; Peter R Møller, Resources, Investigation; Do TT Huong, Nguyen T Phuong, Horst Wilkens, Resources, Writing;review and editing; Annemarie Brüel, Resources, Funding acquisition; Eric Warrant, Writing;original draft, Writing;review and editing; Tobias Wang, Conceptualization, Resources, Supervision, Funding acquisition, Methodology, Writing;original draft, Project administration, Writing;review and editing; Jens R Nyengaard, Conceptualization, Resources, Software, Supervision, Funding acquisition, Methodology, Writing;original draft, Project administration, Writing;review and editing; Michael Berenbrink, Conceptualization, Resources, Supervision, Funding acquisition, Investigation, Methodology, Writing;original draft, Project administration, Writing;review and editing; Mark Bayley, Conceptualization, Resources, Data curation, Supervision, Funding acquisition, Methodology, Writing;original draft, Project administration, Writing;review and editing

## Author ORCIDs

Christian Damsgaard (iD) https://orcid.org/0000-0002-5722-4246
H William Detrich III (iD) https://orcid.org/0000-0002-0783-4505

## Ethics

Human subjects: The Central Denmark Region Committees on Health Research Ethics approved the use of histological sections from human autopsies, project ID M-2012-299-12, file 48447.
Animal experimentation: The ultrasonography was performed with permission from the Danish Inspectorate for Animal Experimentation within the Danish Ministry of Food, Agriculture and Fisheries, Danish Veterinary and Food Administration (Permit number 2016-15-0201-00835). The University of Oregon Institutional Animal Care and Use Committee approved the capture and sampling of notothenioid specimens (Animal Welfare Assurance Number A‐3009‐01, IACUC protocol 12‐02RA).

## Decision letter and Author response

Decision letter https://doi.org/10.7554/eLife.52153.sa1

# Additional files

## Supplementary files

• Supplementary file 1. Number of species included in the data analysis for each physiological or anatomical trait. Most data points were generated within this study, but literature data on maximal retinal thickness and Root effect magnitude as well as literature observations on different capillary types was included in the analysis.

• Supplementary file 2. Concentration of benzocaine to achieve anaesthesia, ultrasound transducer frequency, and resolution of micro-CT. CT, computed tomography; n, numbers of replicates for ultrasound; NA, not available; US, ultrasound. Values are means ± standard deviations.

• Supplementary file 3. Coefficient of variance for retinal thickness, volume of retina and volume of eyes in species used in micro-ultrasound. The coefficient of variance (CV) was calculated as the ratio between standard deviation and sample mean. Values are percentages. CRM, choroid rete mirabile; n, number of replicates for ultrasound CV analysis; T, thickness; V, Volume.

• Supplementary file 4. Interactive 3D model. Three-dimensionally rendered interactive model generated from micro-computed tomography imaging of a goldfish after an arterial injection of a $BaSO_4$-based contrast agent. The interactive file should be viewed in Adobe Acrobat Readernineor higher. To activate the 3D feature, click the model. Using the cursor, it is possible to rotate, zoom, and pan the model. All segments of the model can be turned on/off or made transparent. The model tree is a hierarchy containing several sublayers that can be opened (+). Pre-defined views similar to *Figure 3—figure supplement 1A and C* can be selected below the model tree

• Transparent reporting form

## Data availability

The full raw data set, computer code, and evolutionary trajectories to all species in Figure 6 are deposited on GitHub (https://github.com/christiandamsgaard/Retinaevolution; copy archived at https://github.com/elifesciences-publications/Retinaevolution). All histology-, CT-, ultrasound-, and MRI-scans (~2 TB) can be accessed at https://retinaevolution.bios.au.dk/eLife%20documentation/README.txt.

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
