## [Decision Letter]

**Acceptance summary:**

This study provides insight into the evolution of mechanisms of oxygen delivery to the retina, a tissue with a high metabolic demand. A thicker retina is typically associated with better vision, in most cases due to an increase in the photoreceptor density, which provides for greater resolution (higher acuity), due to an increase in the sampling of photons arriving at the photoreceptor layer. As well, an increase in the density of retinal ganglion cells, the output cells of the retina which transmit the results of the processing of the visual signals, to the brain. These increases cause an increase in retinal thickness, and a concomitant demand for an increase in energy, which can be enabled by an increase in oxygen supply. Oxygen can be supplied to the retina via several mechanisms, which are explored in this study of bony fishes and mammals. Intraretinal vasculature as well as a plexus of vessels, the choroid, adjacent to the support cells of the retina (the retinal pigment epithelium), are the two vascular tissues that can increase or decrease in accord with changes in retinal thickness. In addition, the "Root effect", a mechanism for increasing oxygen delivery via an effect of pH on the affinity of hemoglobin for oxygen, can change the availability of oxygen for the retina. Through an extensive analysis of these parameters, and retinal thickness, the data support a model whereby oxygen supply co-evolved with retinal thickness. This is a very well supported model that provides an excellent example of how a study of physiological parameters can inform our understanding of the evolution of an important trait.

**Decision letter after peer review:**

[Editors’ note: minor issues and corrections have not been included, so there is not an accompanying Author response.]

Reviewer #1:

This tour de force study uses a combination of biochemical physiology, micro-anatomy, and phylogenetic reconstructions to elucidate the evolution of retinal morphology in bony vertebrates, and associated mechanisms of retinal oxygen supply. The experiments are rigorous, the statistical methods are sound, the sample sizes are impressive, the inferences regarding evolved changes in phenotype are well-supported, and the conclusions are biologically fascinating and unexpected.

Overall, this is an extremely comprehensive, biologically rich study that yields fascinating insights into the evolution of high-acuity vision, and it illustrates how changes in one particular sensory system require modifications of physiological phenotypes that seem completely unrelated. For example, Damsgaard and colleagues convincingly demonstrate that retinal morphology of fishes is intimately tied to the manner in which the oxygenation properties of hemoglobin are modulated by changes in red blood cell pH - a fascinating and unexpected result.

The authors' inferences about physiological/morphological trade-offs are well-supported by the phylogenetic patterns of trait coevolution, and they are further bolstered by unique exceptions that prove the rule. For example, to clinch the association between retinal morphology and 'Root effect' hemoglobin, the authors examined Antarctic ice fishes - the only vertebrates that do not possess red blood cell hemoglobin - and documented that the absence of a mechanism for ocular oxygen secretion via the Root effect was compensated by extensive pre-retinal capillarization to sustain oxygen diffusion. They also examined different ecomorphs of the cave-dwelling Mexican tetra that represent different stages of ocular regression. Among these different ectomorphs - which appear to represent different degrees of commitment to a subterranean lifestyle - those that exhibited the most advanced stages of eye regression also exhibited reductions in the associated micro-vasculature that provides a means of counter-current oxygen-exchange (the rete mirabile) and they possessed hemoglobins with the smallest Root effect. The different lines of evidence all point in the same direction and clearly demonstrate that retinal thickness is closely associated with oxygen-supply. The results also highlight interesting compensatory mechanisms for ensuring an uninterrupted oxygen supply.

In summary:

This is an impressively integrative and rigorous study that presents well-supported conclusions and tells a biologically fascinating story. In my opinion, this study is definitely appropriate for a top-tier journal like *eLife*.

Once this paper is published, I would love to add it to the set of papers we discuss in my Evolution graduate course. It nicely illustrates the power of integrating experimental biology with comparative phylogenetic methods and it demonstrates how mechanistic insights into how animals work can shed light on broad-scale patterns of organismal evolution.

Reviewer #2:

I thoroughly enjoyed reading this manuscript. It is a good example on how evolutionary questions can (only) be solved with thorough physiological experimentation. Unfortunately, this view is not yet shared by many evolutionary biologists, but I think that articles like this should make them to see the strength and importance of evolutionary physiology. The work was well written and convincingly argued, so I was left with virtually no criticisms - and this seldom is the case for my reviews.

There is actually only one point, which I think the authors may want to consider. They discuss the reasons for the loss of Root effect Hb and regressed choroid *rete* in the Introduction. While the reasoning is quite convincing for this, they do not address the converse point: many bony fish have evolved and retained Root effect Hb and choroid *rete* instead of evolving choroid capillarization, to me this suggests that the Root effect Hb may be useful also for other purposes than choroid oxygen secretion.